

# Zero-shot reranking with dense encoder models for news background linking

Marwa Essam and Tamer Elsayed

College of Engineering, Qatar University, Doha, Qatar

## ABSTRACT

News background linking is the problem of finding useful links to resources that provide contextual background information for a given news article. Many systems were proposed to address this problem. Yet, the most effective and reproducible method, to date, used the entire input article as a search query to retrieve the background links by sparse retrieval. While being effective, that method is still far from being optimal. Furthermore, it only leverages the lexical matching signal between the input article and the candidate background links. Nevertheless, intuitively, there may exist resources with useful background information that do not lexically overlap with the input article's vocabulary. While many studies proposed systems that adopt semantic matching for addressing news background linking, none were able to outperform the simple lexical-based matching method. In this paper, we investigate multiple methods to integrate both the lexical and semantic relevance signals for better reranking of candidate background links. To represent news articles in the semantic space, we compare multiple Transformer-based encoder models in a zero-shot setting without the need for any labeled data. Our results show that using a hierarchical aggregation of sentence-level representations generates a good semantic representation of news articles, which is then integrated with lexical matching to achieve a new state-of-the-art solution for the problem. We further show that a significant performance improvement is potentially attainable if the degree by which a semantic relevance signal is needed is accurately predicted per input article.

## INTRODUCTION

News background linking is the problem of finding useful background resources that a reader of a given query article can read next for a better contextual understanding of the article. This problem was introduced and addressed mainly in the Text REtrieval Conference (TREC) during the years 2018–2021 (*Soboroff, Huang & Harman, 2018*; *Soboroff, Huang & Harman, 2019a*; *Soboroff, Huang & Harman, 2020*).[1]

Several methods were proposed to address this problem. The most effective one to date (*Soboroff, Huang & Harman, 2018*; *Soboroff, Huang & Harman, 2019a*; *Soboroff, Huang & Harman, 2020*) used the whole query article as a search query in an ad-hoc setting to retrieve the background links. While being effective, this method is still far from

Corresponding author
Marwa Essam, me1709534@qu.edu.qa

[1] The overview paper of 2021 was not released by TREC. However participant papers can be found at https://trec.nist.gov.

being optimal. Furthermore, it depends only on the lexical matching signals in finding the relevant background links. However, in order to find useful background information, one must not only focus on term matching with the query article. Some relevant background articles might be written with a different vocabulary than the query article. Moreover, we hypothesize that for some query articles that do not report specific news events but rather discuss general subjects, deviating from the query article's exact vocabulary is essential in order to find new background links that are not just duplicating the knowledge in the query article. In other words, we need to go beyond the surface-level information that exists in the query article in order to find what actually benefits the reader.

Although some research studies adopted semantic matching in news background linking (*Khloponin & Kosseim, 2019*; *Day, Worley & Allison, 2020*; *Rahul Gautam, 2020*; *Khloponin & Kosseim, 2021*), none showed significant improvement over the simple lexical matching between the query article and the candidate link. Accordingly, in this work, we propose to integrate both the lexical and semantic relevance signals for more effective news background linking. As training data is relatively scarce for the task, we opt to adopt a *zero-shot* setup that leverages the transfer learning capabilities of existing pre-trained Transformer-based encoders for effectively representing spans of text in the semantic space. We propose to split the query article into multiple passages (potentially capturing its subtopics), encode each passage and match it with the candidate background link, and finally aggregate the matching scores for better semantic ranking.

To work in the semantic space, many pre-trained Transformer-based models were recently proposed (*Reimers & Gurevych, 2019*; *Manzil et al., 2020*; *Beltagy, Peters & Cohan, 2020*; *Nishikawa et al. 2022*), each of which has a restricted length of the input sequence and was trained using different text types. While some were trained with sentences as input, for example, others were trained on longer sequences, such as paragraphs, Wikipedia articles, or full news articles. Those models were shown to be effective in different language understanding tasks, such as text classification and document retrieval (*Tay et al., 2020*), page linking (*Yasunaga, Leskovec & Liang, 2022*), sentence similarity and sentence clustering (*Nishikawa et al., 2022*; *Jiang, Zhang & Wang, 2022*). However, to our knowledge, they were not effectively adopted for the specific task of news background linking before. Moreover, since the input context length of most of those models cannot fit both the relatively long query news article as well as its candidate background link, fine-tuning those models for background linking may not be feasible. Accordingly, we aim to study the effectiveness of those models in news background linking, in a zero-shot setting.

Overall, in this work we aim to address the following research questions:

1. **RQ1:** Given the input sequence length restrictions of existing pre-trained encoder models, what is the best model to use, without the need for further pre-training or fine-tuning, to represent news articles for semantic-based news background linking?
2. **RQ2:** How can we integrate both the semantic and lexical relevance signals for an effective reranking of the candidate background links?
3. **RQ3:** Do the query articles vary in their need for the lexical and semantic relevance signals for reranking the candidate background links?

To address the above questions, we propose a *zero-shot* approach that does not need any labeled data. We rely on the recent pre-trained dense encoder models to semantically represent the news articles. For a given query article, we then apply semantic matching to rerank a candidate set of background links obtained initially by sparse retrieval. Our results show that the best dense representation of news articles for our problem is a sentence-level hierarchical aggregation using a sentence-based encoder. Regardless of the representation model though, we found that an effective reranking of the candidate links cannot be achieved solely by semantic matching; we show that a normalized sum of the semantic and lexical matching scores can be used to better rerank the candidate set. This simple zero-shot integration of relevance signals exhibited a statistically significant improvement over the state of the art (SOTA) approach for the news background linking problem. Our analysis of the different query articles further revealed that the degree by which a semantic relevance signal is used should be query-based. If this degree is accurately predicted, further significant performance improvement can be achieved.

Our contribution in this paper is four-fold:

- We present a comparative study between different SOTA pre-trained encoder models for the purpose of document representation in a new downstream task, for which they were not previously evaluated.
- We propose a *zero-shot* reranking technique that integrates lexical and semantic relevance signals for the news background linking task. Our proposed approach exhibits a statistically significant improvement over the SOTA for our task.
- We are the first, up to our knowledge, to conduct experiments and report performance over *all* query sets released by TREC during the four years of the news background linking track, making our results more reliable and conclusions more robust.
- We publicly release the source code of our methods and experiments to support future research on this problem (https://github.com/Marwa-Essam81/ZShotNewsBackLinking).

This paper is organized as follows: First, we review the literature for news background linking, and briefly highlight the work on semantic-based long documents retrieval. We then formally define the background linking problem and distinguish it from other similar problems in the literature. Subsequently, we present our proposed methodology in detail, including the pre-trained models we adopted to transform the news articles to the semantic space, along with a description of how we obtain a semantic representation for a news article given the input length restriction of each model. Next, we show our experimental setup and results, as well as a detailed discussion of the answers to our research questions. We later show the implications of our work, and finally, conclude our work and present our future research directions.

## RELATED WORK

In this section, we first review the work done specifically for news background linking adopting either unsupervised or supervised methods. Then, we briefly outline the lines of work in literature that aim to rank long documents given a search query.

## Unsupervised news background linking

The news background linking problem can simply be addressed following an ad-hoc search approach, in which a search query is extracted from the input query article and an inverted index of a collection of resources is looked up for candidate links that match the search query. Several researchers followed this approach and created search queries using, to name some, the first 1,000 article terms or the ones that have the highest *TF-IDF* values (*Yang & Lin, 2018*), the article's title (*Lopez-Ubeda et al., 2018*) or its full content (*Bimantara et al., 2018b*; *Lu & Fang, 2019*; *Khloponin & Kosseim, 2020*; *Engelmann & Schaer, 2021*), with some methods adopting clustering techniques to diversify the results for the news reader. Several teams focused on extracting the most representative keywords, or sometimes named entities, from the query article to be used as the search query (*Bimantara et al., 2018a*; *Lua & Fang, 2018*; *Lu & Fang, 2019*; *Essam & Elsayed, 2019*). Expansion of the search queries using Rocchio and RM3 was also experimented with to avoid the exact match problem (*Missaoui et al., 2019*; *Ding et al., 2019*).

Another unsupervised background linking method was proposed that converts each article to a node graph, and applies similarity measures between the query article graph and other graphs to obtain the ranked list of background links (*Wagenpfeil, Mc Kevitt & Hemmje, 2021*). A formula was proposed further that scores each background link given the similarity between the named entities mentioned in this link and the ones mentioned in the query article (*Engelmann & Schaer, 2021*). The parameters of this formula were selected based on the analysis of the query articles released in 2020 and its relevant background links. This technique slightly outperformed the author's implementation of the full article retrieval approach on the 2021 set of query articles. Yet, the authors showed that it did this on only a small number of query articles with no overall significant change in performance.

Another technique was proposed (*Zhang et al., 2022*) that attempts to selectively expand the query articles during the lexical matching process based on whether or not this article is reporting time-sensitive information. To retrieve the candidate background articles though, the authors used, as a search query, the description of what readers want from a background article (*i.e.,* the information need). This description was only available for the TREC 2021 set of query articles that the authors experimented on, and was given by the track organizers in TREC that year to show participants examples of what a reader would want when seeking background articles. However, in a real life scenario, this description is not intuitively known ahead, as news providers give readers only the news articles without a description of what is missing in it (required background knowledge). Therefore, it is the goal of a background linking system to discover what is missing and use it to retrieve the required background links.

Most of the unsupervised techniques shown above, in one way or another, attempted to capture a sort of lexical similarity or distance between the query article and its background links. While important, some useful background links may be contextually helpful, yet written with different vocabulary, different writing style, or using different entities, than the query article. Hence, we hypothesize that ensuring that the semantics of both articles are captured and evaluated is further essential to ascertain that all useful background articles are retrieved during the background linking process.

## Supervised models for news background linking

Some researchers attempted to harness the capabilities of supervised models for news background linking. For instance, using different features extracted from both the query articles and the candidate articles (such as its textual similarity, difference in publication dates, candidate document length, clickbait-probability of candidate title, *etc.*), some methods were proposed to learn a reranking model for the candidate background links (*Foley, Montoly & Pena, 2019*; *Qu & Wang, 2019*; *Koster & Foley, 2021*). Another background linking method further trained a LambdaMART model post converting articles into a graph of named entities and extracting the training features from the graph (*Ornella & Silvello, 2021*).

A couple of news background linking methods adopted BERT (*Devlin et al., 2018*) for background linking (*Ak et al., 2020*). The first used a BERT-based extractive summarization model to generate a summary of 180 words from the query news article, then used this summary as a search query to retrieve the background links. The second method fine-tuned the next sentence prediction task in BERT for background linking. Since BERT input is limited to 512 tokens, both the query article and the candidate background article were cropped to 256 tokens during the fine-tuning process.

Some methods were proposed that are based on representing articles in the semantic space, then matching the query article's representation to the ones of the candidate background articles using similarity measures, with sometimes aggregating the semantic matching scores with other relevance signals. Among the models adopted for news articles representation was the Word2vec model (*Church, 2017*). Post training the model on the news articles collection (*Rahul Gautam, 2020*), articles were represented as the sum of the vectors of its most frequent 25 words. Other pre-trained models were also experimented with without further training on the news articles collection. The models experimented with were: Doc2Vec (*Khloponin & Kosseim, 2019*), Sentence-BERT (*Reimers & Gurevych, 2019*) using the pooled vectors of the first three paragraphs of the article (*Day, Worley & Allison, 2020*), GPT2 post splitting the article into 250 tokens chunks and mean-pooling the chunks representations (*Khloponin & Kosseim, 2021*), LDA to obtain the topic representation vectors of the articles (*Ajnadkar et al., 2021*), Sentence-BERT using the representation of keywords extracted from the article (*Sethi & Deshmukh, 2021*), and Sentence-BERT using an average over all tokens representations from the article (*Cabrera-Diego, Boros & Doucet, 2022*).

While the work reviewed above attempted to capture the semantics of the query and the candidate background links for a better linking process, the techniques proposed treated the query article as one entity. Most often though, a news article discusses several subtopics related to its main topic within its paragraphs, and a useful background article may extend the reader's knowledge on just one or more of these subtopics, not necessarily all of them. Accordingly, we hypothesize that an effective background linking technique needs to evaluate the information provided in the candidate background article's on each subtopic in the query article, then aggregate the results across subtopics. In the next section, we will explain in detail our proposed work to validate this hypothesis.

### Long documents ranking

There is an ongoing research for matching long documents with a search query based on its semantics, such as the work done for question-answering (*Nguyen, MacAvaney & Yates, 2023*; *Rau, Dehghani & Kamps, 2024*). However, in this line of research, the search query is always relatively short and can be easily represented by different language models. Hence, the challenge becomes on how to represent the long candidate documents during the semantic matching process. This is in contrast to news background linking, where both relatively long documents need to be evaluated for background relevance. Some recent work focused on proposing models for matching pairs of long documents for other tasks such as finding scientific articles with common citations (*Pang, Lan & Cheng, 2021*; *Jha et al., 2023*). Yet, as far as we know, the checkpoints for these trained models have not been publicly released, and they need to be trained from scratch for either inference or fine-tuning purposes. In this work though, we opted to experiment with the language models that have available open source encoders as we will show later.

Finally, it is important to highlight here that, considering the literature review for all techniques proposed to address the news background linking problem, the most effective and reproducible method proposed to find the background links was a brute-force method that used the full input query article as a search query in an ad-hoc setting to retrieve the background links. This method was introduced within TREC by different teams using different platforms and with different preprocessing setups. Yet, regardless of the setup, it still outperformed other ideas, making it the SOTA for the news background linking problem. Accordingly, we focused in our experiments on the comparison with solely this later method, as we will discuss later.

## BACKGROUND LINKING PROBLEM

In this section, we define the news background linking problem, distinguish it from other similar problems in the field, then we provide examples that highlight its difference with those other problems.

### Problem definition

Let us start with a user scenario. A news reader is reading a news article $q$, but does not fully comprehend it. The reader may seek other informative background sources $B$ that provide more context and background knowledge required to comprehend the content of the article, from a pool of external resources $C$. Figure 1 illustrates that scenario.

While a background resource or link can be any document with useful information to the reader, in this work, we focus on the specific case where the background resources or links are other news articles from the news collection of $q$, which is followed in TREC.

### Related problems

Finding *related* articles in general has been studied in the literature for many years. News recommendation is the closest problem in definition to background linking, assuming that we search a collection of only news articles (*Karimi, Jannach & Jugovac, 2018*). The difference though between the two problems is that in news recommendation,

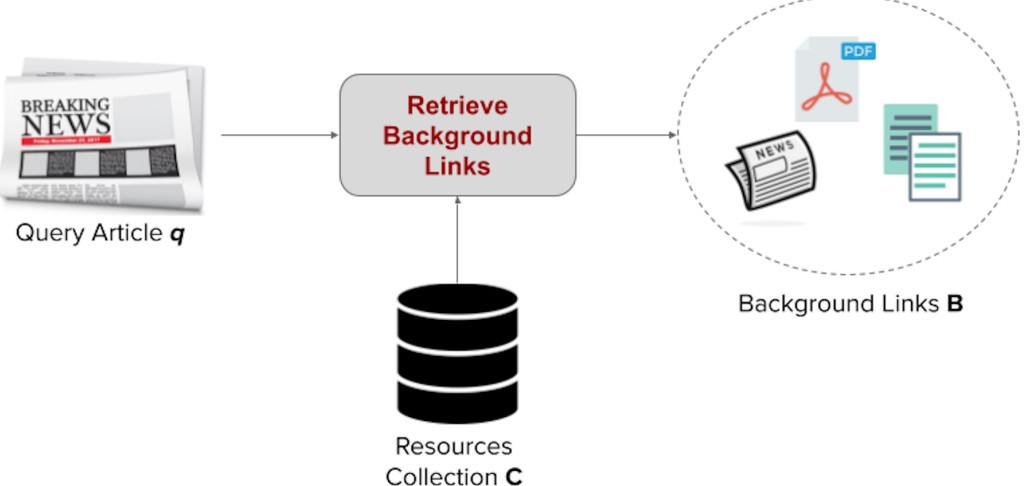

**Figure 1** The news background linking problem.

the proposed system uses the readers profile including their browsing history, online interactions, or sometimes their location during the recommendation process. This usually helps the recommendation system to recommend news articles that are of interest to the reader (*Zhang, 2023*; *Tian et al., 2021*; *Hu et al., 2020*; *Ma et al., 2022*). However, in news background linking, readers are assumed to be anonymous, with no possible way of finding any information related to them. Accordingly, the attention is mainly on the content of the query article. Some studies (*Salih & Jacksi, 2020*; *Sarwar, Noor & Miah, 2022*) attempted to recommend articles to anonymous readers if it discuss the same main topic as the current article they are reading. Yet, articles that generally discuss the same topic may not necessarily provide background knowledge to one another.

There has been some further work recently on event detection within new articles (*Rospocher et al., 2016*; *Qian et al., 2019*; *Silvano et al., 2024*), and also on analyzing or finding the storylines of those events (*Nicholls & Bright, 2019*; *Campos et al., 2021*; *Zhuang, Fei & Hu, 2023*). In most of that work though, an event is assumed to definitely exist within the news article, and the goal of the system is to extract the components of this event (what happened, who, when, where, and how), or to find the temporal relations between the mentioned events. This is not assumed in background linking, as some articles may discuss general subjects and may not at all refer to specific events (*e.g.*, a feature article that discusses the high fees of college education). Additionally, in some cases, a background article may be helpful to the reader even though it does not mention the specific event of the query article.

## Examples

Figure 2A shows an example of news background linking, where the query article discusses the progress in the trial of a truck driver who was accused of the death of 10 illegal Mexican immigrants in his truck. The relevant background articles give the reader more context and background information on this incident. One article for example discusses how the

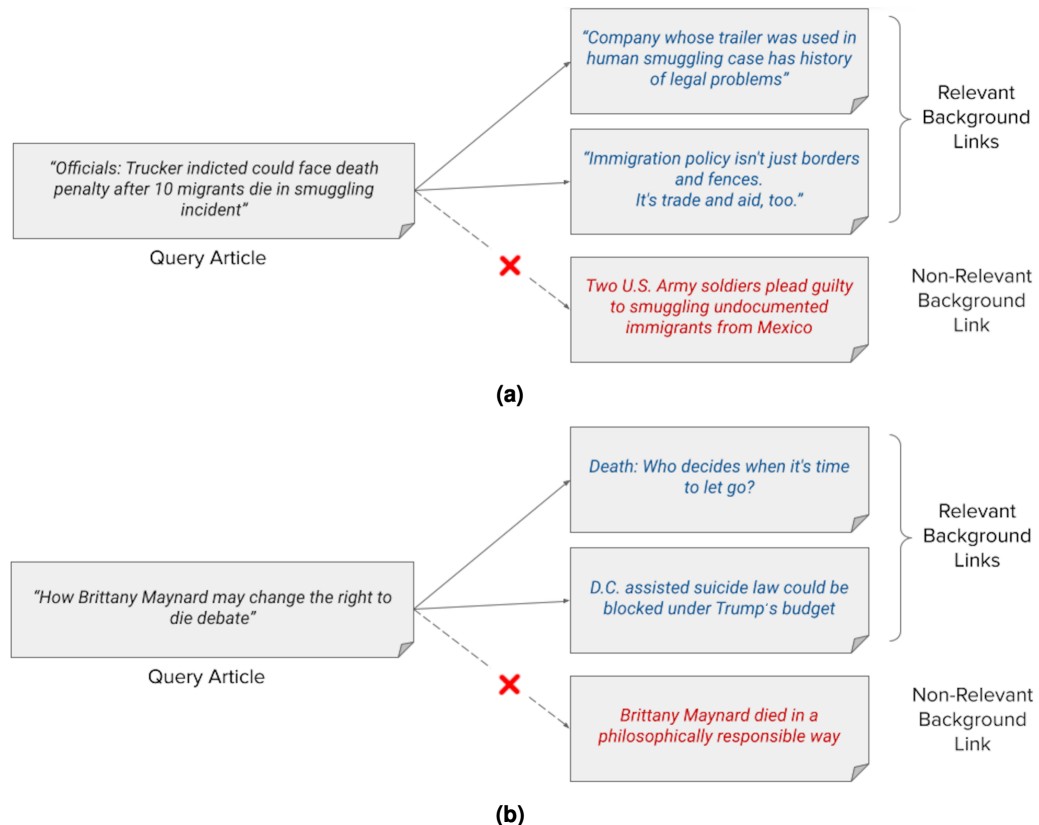

**Figure 2** (A–B) Examples of news background linking drawn from the TREC dataset for the news track.

company that owns the truck, for which the driver works, is famous for falsifying records. The second article does not mention this specific incident, however it discusses the patterns of Mexican immigration to the United States, and how the immigration pool should be handled. The non-relevant article in Fig. 2A also discusses a case of illegal immigrants from Mexico, but it does not at all contain information that helps readers of the query article understand its content or even contextualize it. Instead, it just details the possible consequences that two soldiers will face after being convicted for illegally driving two Mexican men through an immigration checkpoint. It is important to note here that a typical news recommendation system may retrieve this specific article and link it to the query article as it might be of interest to a reader who reads about illegal immigration cases to the US for example or the world in general.

Another example is shown in Fig. 2B. The query article discusses the effects of the death of a young lady "Maynard", who took pills to die after being diagnosed with cancer, on the "right to die" debate in the United States. The background articles give the reader more context on this debate as it is the main context of the query article, without even mentioning the specific case of "Maynard". The first article discusses the approval of the "right to die" law in different states after case courts. The second relevant article discusses

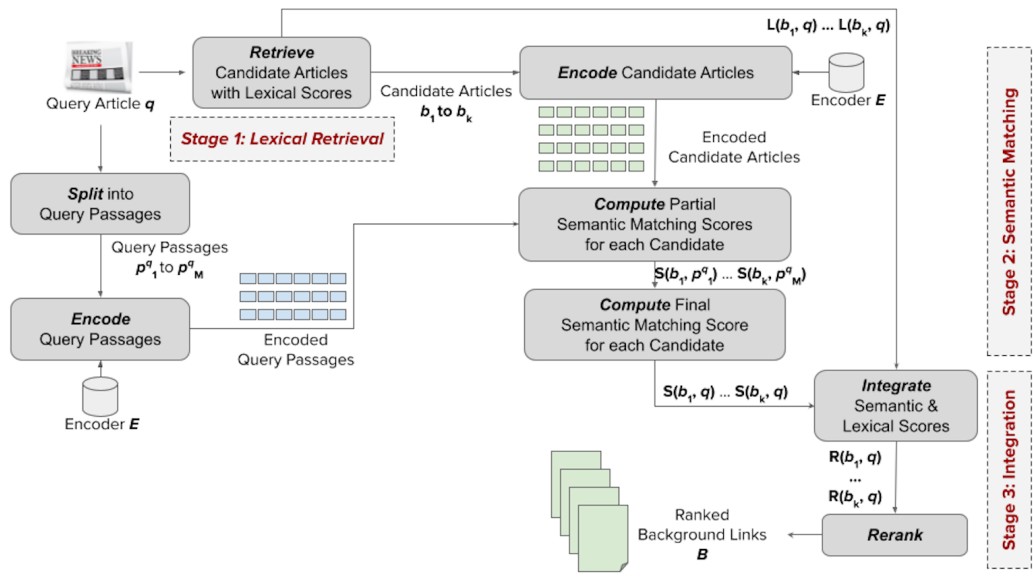

**Figure 3  Our proposed framework for addressing the news background linking problem.**

an economic view on approving the right-to-die law in former President Trump's budget. The non-relevant background article, while mentioning the case of "Maynard", merely talks about how people like "Maynard" who devoted their few remaining days in life to a cause are not self-centered. It is important to note here further that a system that aims to find storylines may link the query article to this former non-relevant article as both mention the specific case of "Maynard's" death. It will further exclude the other relevant background links from the results.

## PROPOSED FRAMEWORK

In this section, we present our proposed framework that addresses the problem of background linking of news articles through the integration of both the lexical and semantic relevance signals between an input query article and its candidate background links. We start by describing the general structure of this proposed framework, then we delve into details on how this framework actually works.

Figure 3 shows the general structure of our proposed framework. It takes an input query article $q$ for which we want to retrieve useful background links, and eventually provide a ranked list $B$ of $k$ background articles. The framework has three stages: (1) lexical retrieval, (2) semantic matching, and (3) integration of lexical and semantic signals.

At the first stage, denoted by *Lexical Retrieval*, we reduce the space of potential background links by retrieving, from the news articles collection, a candidate list $B$ of $k$ news articles having the highest *lexical* similarity (denoted as $L(b, q)$) of a candidate article $b$ to the query article $q$. This is done using a basic sparse retrieval model (*e.g.*, BM25). At the second stage, denoted by *Semantic Matching*, we *semantically* match the input article $q$ against each candidate $b$ to obtain semantic matching scores. To achieve that,

we first split $q$ into $m$ query passages $p^q_{1..m}$. Using a Transformer-based encoder model $E$, we then encode each passage $p_i$ and each candidate article $b$ into embedding vectors. Next, we compute semantic matching scores between each candidate article and each passage, denoted as $S(b, p^q_i)$. The passage matching scores for each candidate are then aggregated into a semantic matching score $S(b, q)$. Finally, at the *Integration* stage, the lexical and semantic matching scores are combined into a final score $R(b, q)$ that is used to rerank the candidate background articles.

Note that the framework proposes a zero-shot reranking that does not need any labeled data; it counts on the pre-training of the encoder to represent the candidate links in the semantic space and integrates the lexical (sparse) and semantic (dense) retrieval scores in an unsupervised manner. We discuss the details of all stages of the framework in this section.

## Stage 1: lexical retrieval

This stage is quite simple. It aims to obtain a *candidate* list of news articles that most probably contain relevant and useful background links. For this, the text of each news article in the given articles collection is initially preprocessed (lowercased and stop words are removed), then it is indexed into an inverted index. For a given query article, the title of the article is concatenated with its text to compose a search query, which is preprocessed before being issued. Any robust sparse retrieval model (such as BM25; *Robertson et al., 1996*) can be used to retrieve the candidate background links.

## Stage 2: semantic matching

In the semantic matching stage, we first split the query article into shorter passages, potentially capturing its subtopics. Then we encode each passage and each candidate article into the semantic space. Finally, we match each candidate against each query passage and aggregate the passage scores per candidate to obtain a candidate score. We detail all of these steps in this section.

### *Splitting query article into passages*

As discussed earlier, the goal of this stage is to match each candidate background article semantically against the query article. We observed that the query article might have multiple subtopics and the candidate article might have the background knowledge or provide the context for one or more of them. Accordingly, we propose to split the query article into a number of shorter passages, called query passages, aiming for each passage to represent a subtopic. Then we semantically match each query passage with the candidate article. We hypothesize that a subtopic is addressed in a few consecutive paragraphs. Hence, we split the article into individual paragraphs, and use a sliding window of two consecutive paragraphs (with a step size of one paragraph) to construct query passages that each, hypothetically, represents a subtopic. A query passage is the concatenation of the enclosed paragraphs within the sliding window. We consider the title of the article as the first paragraph.

**Table 1  Pre-trained encoder models used in our experiments along with the type of input they were pre-trained on.**

| Model | Encoder source | Input |
|---|---|---|
| *SBERT* | sentence-Transformers/all-mpnet-base-v2 | Sentence |
| *EASE* | sosuke/ease-roberta-base | Sentence |
| *PromCSE* | sup-PromCSE-RoBERTa-large | Sentence |
| *LinkBERT* | michiyasunaga /LinkBERT-base | Paragraph |
| *Ernie-2.0* | nghuyong/ernie-2.0-base-en | Paragraph |
| *BigBird* | google/bigbird-roberta-base | Long text |
| *LongFormer* | allenai/longformer-base-4096 | Long text |

### Contextual representation of news text

The next step of our proposed approach is to obtain a dense representation of both the query passages and the candidate background articles. Due to limited labeled data available for the news background linking problem, we opted to leverage existing Transformer-based encoder models that were pre-trained for several natural language processing (NLP) tasks in a zero-shot setting. Hoping to have a good representation of the news articles using these models, we set the following criteria when selecting the models to experiment with:

- Models that were pre-trained for text contextualization on datasets that included news articles (*e.g.*, Sentence-BERT and BigBird).
- Models that were fined-tuned on pre-trained language models to learn a representation of text that included links to other text sources (such as hyperlinks between documents in Wikipedia), counting on the closeness of this task to news linking (*e.g.*, LinkBERT).
- Models that were pre-trained to learn representation of text that included named entities, since news articles often include references to persons, organizations, or locations (*e.g.*, EASE and ERNIE-2.0).
- Models that were shown to be effective for tasks that depend on acquiring a good low-dimensional representation for long documents, such as document matching and document classification (*e.g.*, LongFormer).
- Models that showed effective performance when evaluated on out of domain tasks (*e.g.*, PromCSE).
- Models that have available open-source pre-trained encoders.

Given the above selection criteria, we further categorized the models into three categories based on the length of their training samples. We experiment with models that were trained on sentence-level input, limited-size input (512 tokens), and long-text input (up to 4,096 tokens), as shown in Table 1.

Our goal is to test the ability of those models in capturing the core context presented within each candidate news article to allow for effective news background linking. In other words, a good model should capture the valuable information reported in an article and hence provide a representation that enables effective semantic matching between the query article and its relevant background links. We use those encoder models in a *zero-shot* setting; no labeled data for news background linking was used to further pre-train or fine-tune any

model. This allowed us to test the ability of each model to derive a representation for news articles that can be used for effective new background linking, without specifically being trained for this task.

Since some encoders restrict its input context to a limited number of tokens, feeding a relatively long news article as is to the encoder will often result in the truncation of big parts from the article, leading to the loss of important contextual information. Accordingly, we propose to split the news article into small pieces of text that each is fed independently to the encoding model, and then aggregate the obtained representations to get the final dense vector of the long article. News articles are often written as a number of paragraphs where the lead paragraph shows the most important information about the reported topic or event and the other paragraphs show supporting details or evidences. For simplicity and using this semi-structured nature of news articles, we propose to split the article into its own composing paragraphs and sentences while feeding it to the encoding models that either has input length limits or were trained using limited-size inputs. Following, we discuss briefly how each model we selected was pre-trained, and how we use it using our proposed splitting strategy to obtain a dense representation for both the query passages as well as the candidate articles.

### Models trained on short-text input (sentences):

- **Sentence-BERT** (SBERT) (*Reimers & Gurevych, 2019*) was proposed as an extension of BERT (*Devlin et al., 2018*) to derive semantically meaningful sentence embeddings. Recently, Sentence Transformers have been proposed as a family of **SBERT** models as encoders for short text (sentences). Each of those models was trained on a different dataset from a different domain. The model we experiment with in this study, all-mpnet-base-v2, was fine-tuned from the pre-trained microsoft/mpnet-base model on a dataset of 1 billion sentence pairs with a self-supervised contrastive learning objective that aimed to identify pairs of contextually similar sentences. As suggested by the framework, we obtain the sentence embeddings using this model through the average pooling of the sentence token embeddings.

- **EASE** (*Nishikawa et al., 2022*) similarly aims to capture the semantics of sentences, while specifically focusing on identifying the information of related entities in sentences. Wikipedia sentences with hyperlinks treated as related entities were used for training this model. Similar to SBERT, we obtain the sentence embeddings through the average pooling of the sentence token embeddings.

- **PromCSE** (*Jiang, Zhang & Wang, 2022*) aims to overcome the drop in performance faced when sentence encoders are evaluated on out-of-domain tasks. To do that, it prepends soft prompts (limited sequence of continuous vectors) at each layer of the pre-trained language model (BERT or ROBERTA) during the fine-tuning process, while freezing all the other model parameters. Since it showed SOTA performance when evaluated in new domains, we opted to experiment with it for news background linking. As recommended (https://github.com/YJiangcm/PromCSE), we obtain a sentence embedding from the [CLS] token hidden state from the last layer of the model.

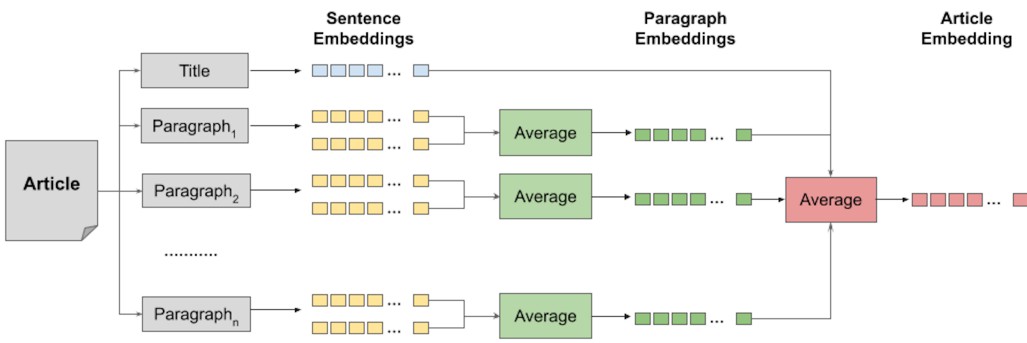

**Figure 4** Sentence-based hierarchical dense representation of news articles using Transformer-based encoders.

To obtain a representation vector for a candidate news article using the above sentence-based encoders, we first split each paragraph in the article into sentences,[2] encode each sentence independently, then obtain an encoding vector for the paragraph by averaging the embedding vectors of its sentences. Finally, we average all vectors of the paragraphs and title as well to obtain an overall representation for the article. This hierarchical encoding approach is illustrated in Fig. 4. We used the average function here to capture the common associations between the vectors of those elements. We hypothesize, therefore, that the average captures the actual context of the article.

We further adopted this approach to get the vector representation of each passage of the query article. We first obtained representation vectors for the paragraphs within the passage, then we averaged those representation to get the passage representation vector.[3]

### Models trained on limited-size input (maximum 512 tokens):

- **ERNIE-2.0** (*Sun et al., 2020*) is a model trained for natural language understanding tasks through continual multi-task learning. The training tasks relied on weak-supervised signals that were obtained from data such as named entities, adjacent sentences, and discourse relations. It was shown to be effective on many language understanding tasks.[4]

- **LinkBERT** (*Yasunaga, Leskovec & Liang, 2022*) is a BERT-based model trained on Wikipedia by feeding pairs of documents with hyperlink or citation information to the same model to obtain a contextual representation of text that incorporates more knowledge about the links within. While the content of news articles is somehow different, we hypothesize that LinkBERT might be able to similarly capture the semantic links between the query article and its background articles.

We adopt the same hierarchical approach above to obtain a single embedding vector for each candidate article and for the query passages, except that we directly get the embedding vectors for paragraphs using the above models without the need to split them into sentences.

### Models trained on long-text input:

- **BigBird** (*Manzil et al., 2020*) is a Transformer-based model that extends the traditional Transformer-based models to allow longer sequences of input to be processed through

---

[2] We used the Python *segtok* library for this (https://github.com/fnl/segtok.git).

[3] Since the SBERT model was trained on long sentences (up to 128 tokens), we only split a paragraph into sentences if it exceeds that length; otherwise, we feed it to the model as is.

[4] Within the General Language Understanding Evaluation (GLUE) benchmark.

**Peer**J Computer Science

[5] Articles that exceeded 4,096 tokens in length were truncated.

the concept of sparse attention. It also applies global attention on tokens such as the CLS token, as well as random attention. This model is shown to be effective in both document classification and document matching tasks (*Tay et al., 2020*), showing its ability to provide a good representation of long documents.

- **LongFormer** (*Beltagy, Peters & Cohan, 2020*) is a similar and competitive model to Bigbird that uses a combination of sliding window attention and global attention to avoid the quadratic processing time needed for the full-attention mechanism.

Since the above models allow for very long sequences of text input (up to 4096 tokens), we obtain the vector representations of the candidate articles and also the query passages directly using those models.[5]

### Computing semantic scores

After obtaining the vector representation of the query passages and candidate articles, we compute the semantic scores in a straight forward way. Let $E(p_i^q)$ denotes the encoding of the query passage $p_i^q$ and $E(b_j)$ denotes the encoding of the candidate background link $b_j$. A semantic score for a candidate news article $b_j$ can then be computed through either the *average* or *maximum* aggregation functions as follows:

$$S(b_j, q) = \frac{\sum_{i=1}^{m} sim(E(b_j), E(p_i^q))}{m} \qquad (1)$$

$$S(b_j, q) = \max_{i=1}^{m} sim(E(b_j), E(p_i^q)) \qquad (2)$$

where *sim* is the cosine similarity function over the two input embedding vectors.

## Stage 3: integration of lexical and semantic signals

We next integrate both the semantic and lexical matching signals to finally rerank the candidate articles for an effective news background linking. While different query articles might benefit from mixing both relevance signals with different balances, we could not train a model to learn the optimal integration weights due to the very limited number of labeled query articles released for this problem (*i.e.*, insufficient data for training and testing the model). Accordingly, for simplicity, we assume that each relevance signal contributes equally to the final score of a candidate background link. We experimented with two known integration approaches in this regard: score aggregation and rank aggregation, surveyed from the literature (*Depeursinge & Müller, 2010*).

### Score aggregation

We experimented with the following score aggregation functions to compute the final score $R(b_j, q)$ of a candidate news article $b_j$:

- **SumOfScores**: $R(b_j, q) = L(b_j, q) + S(b_j, q)$
- **MaxScore**: $R(b_j, q) = max(L(b_j, q), S(b_j, q))$
- **MinScore**: $R(b_j, q) = min(L(b_j, q), S(b_j, q))$
- **ProductOfScores**: $R(b_j, q) = L(b_j, q) \cdot S(b_j, q)$

<Title>: Teen camper wakes up to 'crunching noise' — and discovers his head is inside bear's mouth'
<Paragraph>: <Sent>Asleep in the mountains northwest of Boulder, Colo., a teenage camp counselor was awakened Sunday morning by a loud crunching sound. <Sent>The noise was the sound of large teeth scraping against his skull, he told Denver ABC-affiliate KMGH.
<Paragraph>:<Sent>A black bear, the teen realized, was trying to pull him out of his sleeping bag by his head. <Sent>"It grabbed me like this and pulled me, and then it bit the back of my head and dragged me," the teen, identified only by his first name, Dylan, told KMGH. <Sent>"When it was dragging me, that was the slowest part. It felt like it went forever."
.....

**Figure 5** An example showing an article split into paragraphs and sentences to prepare it for the encoding process.

- **MixedMinMaxScore**: $R(b_j,q) = max(L(b_j,q), S(b_j,q)) - \frac{min(L(b_j,q),S(b_j,q))^2}{max(L(b_j,q),S(b_j,q))+min(L(b_j,q),S(b_j,q))}$

Before applying the aggregation function, we normalize both the lexical and semantic scores by applying sum-to-unity normalization. This normalization turns the scores into probabilities, such that the sum of scores of candidate articles in a specific ranking for a query is 1.

### Rank aggregation

Let us assume that lexical or semantic scoring assigns a decreasing rank for each candidate news article $b_j$ such as the most relevant article is ranked 1. Let $L_{rank}(b_j,q)$ and $S_{rank}(b_j,q)$ be the rank given to the candidate article $b_j$ after lexical and semantic matching respectively. We applied rank aggregation using either of the following two popular ranking methods:

- **Borda Rank**: $R(b_j,q) = (1 - \frac{L_{rank}(b_j,q)-1}{100}) + (1 - \frac{S_{rank}(b_j,q)-1}{100})$
- **Dowdall Rank**: $R(b_j,q) = \frac{1}{L_{rank}(b_j,q)} + \frac{1}{S_{rank}(b_j,q)_{b_j}}$

### Framework overview

Algorithm 1 shows our proposed news background linking approach. The algorithm takes the query article *queryArticle* as input, the inverted index where the news articles collection is indexed, and the encoder model used for encoding the news articles text. It starts by splitting the query into its own paragraphs, then creates passages using the sliding window method explained before. It then encodes each passage and stores this encoding in a list. Next, the candidate background articles are retrieved (using the lexical similarity matching method), along with their lexical scores. For each candidate article, the candidate is first encoded using the encoder model, then partial scores are computed against the query passages. After obtaining the partial scores for the candidate article, the scores are aggregated to get the candidate's semantic score. Finally, the lexical scores and the semantic scores for candidates are normalized before they are integrated and sorted to produce the final ranked list. For experiments that needed only the semantic scores for ranking, we

skip the integration of signals and return the candidate articles ranked given their semantic scores.

We note that when calling the encoding method with models that take inputs of limited length, it internally implements the hierarchical encoding and aggregation approach explained earlier. Figure 5 shows part of an example news article and how the article's text is split before feeding each piece to the encoder model.

---

**Algorithm 1** Background News Articles Retrieval

---

    **Input** queryArticle, index, encoderModel
    **Output** rankedArticles

1:  **procedure** RETRIEVEANDRERANKARTICLES
2:     $paragraphs \leftarrow splitIntoParagraphs(queryArticle)$
3:     $encodedPassages \leftarrow []$
4:     $i \leftarrow 0$
5:     **do**
6:         $passage \leftarrow concatenate(paragraphs[i], paragraphs[i+1])$
7:         $encodedPassages.append(encode(passage, encoderModel))$
8:         $i \leftarrow i+1$
9:     **while** $i < (length(paragraphs) - 1)$
10:    $candidates, lexScores \leftarrow retrieve(index, queryArticle)$
11:    $semScores \leftarrow []$
12:    **for** candidate in candidates **do**
13:        $encodedCandidate \leftarrow encode(candidate, encoderModel)$
14:        $partialScores \leftarrow []$
15:        **for** e in encodedPassages **do**
16:           $partialScores.append(cosineSim(encodedCandidate, e))$
17:        $semScores[candidate] \leftarrow aggregatePartialScores(partialScores)$
18:    $normLexScores \leftarrow normalize(lexScores)$
19:    $normSemScores \leftarrow normalize(semScores)$
20:    $relScores \leftarrow integrateScores(normLexScores, normSemScores)$
21:    $rankedArticles \leftarrow sortInDescendingOrder(relScores)$

---

# EXPERIMENTAL EVALUATION

In this section, we present our experimental evaluation. We first describe our experimental setup, and then discuss the experimental results that address our research questions. Finally, we briefly state the limitations of our study.

## Experimental setup

Our experimental setup covers the dataset description, the pre-processing and indexing phase, the baseline method, the retrieval and ranking method, and the evaluation measures.

### Dataset

We conducted our experiments on the Washington Post news test collection version 3 released by TREC for the background linking task (https://trec.nist.gov/data/wapost). It covers multiple years of the newspaper publication with about 672k documents comprising news articles, columns, and blogs. There are 50, 57, 49, and 51 query articles (a total of 207) provided by TREC in 2018, 2019, 2020, and 2021, respectively, for the news background linking task. A query article is associated with a set of manually labeled background articles, each of which is assigned a relevance score from 0 to 4 depending on how much context and background knowledge this article provides to the reader of the query article (*Soboroff, Huang & Harman, 2018*; *Soboroff, Huang & Harman, 2019a*; *Soboroff, Huang & Harman, 2020*).

### Preprocessing and indexing

Each article in the dataset is written as a JSON object, that has fields for the article's title and its content paragraphs. We extracted the metadata (title, author, URL, and publishing date) that defines each unique article in the dataset, and concatenated its textual content (marked by "paragraph" type). We then used JSOUP library (https://jsoup.org/) to clear the raw text from the HTML tags. We further lower-cased the text and removed stop words. Finally, the pre-processed text of each article was indexed, along with its meta-data, using Lucene v8 (https://lucene.apache.org/). We did not apply stemming to the text as, in our preliminary experiments with the full-article baseline, the performance was degraded compared to non-stemming.

### Baseline

Since the news background linking problem was primarily addressed within TREC, we elected to choose the baseline as the most (to date) effective method on the TREC test collection. In this method (*Bimantara et al., 2018b*), a search query is constructed out of the concatenation of the query article's title and its body content, then this search query is issued against an inverted index of candidate articles to simply retrieve the background links. This full-article retrieval approach, while being implemented by different research teams across the years, with different environments (retrieval platforms, retrieval models, stop-word lists, stemming options, etc.), showed the highest performance in the 2018, 2019, and 2020 news background linking track runs. In 2021, a system was proposed (*Engelmann & Schaer, 2021*) that showed a slight improvement over its authors' implementation of the baseline method on the 2021 set of query articles, with no significant change in the performance over most of the queries, as shown by the authors. Additionally, the authors' approach was not shown to be effective on unseen queries of earlier years of the news background linking track. Hence, we constitute the *full article* baseline as the SOTA on the background linking problem so far and focus our comparison with only that baseline. Up to our knowledge, there is no implementation available for the baseline method that can be used to obtain unified results on all queries released across all years. Accordingly, we implemented our own version of this method. To make the results from this paper comparable for future research on this problem, we report our computed scores and publicly release our source code and environment setup.

### Initial retrieval and reranking

We used the baseline method to retrieve an initial set of candidate background links for each query article from the constructed index. Within Lucene, we employed the default scoring function (as previously described in *Essam & Elsayed, 2023*) to obtain the lexical matching scores for all the articles in our experiments. This function implements the BM25 retrieval model. While there are other retrieval models available within Lucene, our earlier work with the baseline method showed that the default retrieval model yielded the best ranking results. During the retrieval process, we initially set Lucene's retrieval hit to 1000. Then, upon the retrieval of the candidate articles, we excluded articles that were declared by TREC to be non-relevant such as "Opinions", "Letters to the Editor", and "The Post's View", and articles that were published after the query article. Post this filtering process, we retained a maximum of 100 candidate articles for each query article for reranking. Using the articles' text, *i.e.,* without preprocessing, we obtained the semantic representations for each query passage and each candidate article as detailed before. We then computed the semantic similarity scores required for the reranking process, and finally reranked the retrieved set of articles.

### Evaluation measure

We used the normalized discounted cumulative gain (nDCG) as the evaluation measure to evaluate the effectiveness of the proposed reranking system (*Järvelin & Kekäläinen, 2017*). nDCG was adopted by TREC as it measures the quality of a document recommendation system taking into account the position of the documents in the produced ranked list. This is important as often users/readers explore only the top few recommended articles. We set the ranking depth to 5 (*i.e.,* a system is required to retrieve the top five most relevant background articles in order). Per TREC guidelines, relevance levels were on a scale from 0 (non-relevant candidate article) to 4 (the most relevant). While computing the nDCG metric, each retrieved background article had a gain of $2^r$, where $r$ is the relevance level of that article.

To ensure that the reported performance improvement over the baseline, if any, is statistically-significant, we performed a pairwise $t$-test on the reported performance against the baseline performance with a 95% significance level.

## Results and discussion

Following, we present the experimental results, addressing each of our aforementioned research questions.

### Semantic representation of articles for news background linking

To address **RQ1**, we reranked the candidate background links for each query using only the semantic matching scores achieved by the different models as we presented earlier. This shows the effectiveness of each of those representation models for our task. Table 2 presents the results of this experiment, from which we can draw multiple observations.

First, comparing the performance of the different types of models, the sentence-level models are clearly outperforming the limited-size-based and long-text-based models. More specifically, **SBERT** exhibits the best performance among all models. This can be due to our

**Table 2  nDCG@5 performance of semantic matching using different semantic representation models.**

| Model | Model input | Maximum aggregation | Average aggregation |
|---|---|---|---|
| SBERT | Sentence | 0.4145 | 0.4360 |
| EASE | Sentence | 0.3376 | 0.3862 |
| PromCSE | Sentence | 0.3905 | 0.4032 |
| ERNIE-2.0 | Limited-size | 0.2878 | 0.2921 |
| LinkBERT | Limited-size | 0.3057 | 0.3236 |
| BigBird | Long-text | 0.2059 | 0.1856 |
| LongFormer | Long-text | 0.1473 | 0.1462 |
| Baseline | | | 0.4856 |

fine-grain construction of the hierarchical article representation that captured its semantics. It even achieved better performance than *EASE* and *PromCSE*, which were proposed more recently, and showed high effectiveness on several standard datasets for Semantic Textual Similarity (STS) tasks (*Nishikawa et al., 2022*; *Jiang, Zhang & Wang, 2022*).

Second, surprisingly, the performance of the long-text-based models was the worst, although they were shown to be effective on a number of tasks that included long text matching (*Tay et al., 2020*).

We further notice that average aggregation is generally better than the maximum aggregation in this task. This is somewhat expected for our task, as the average aggregation combines more signals of context background across the query article than the maximum aggregation.

Finally, we notice that our best model (SBERT) is outperformed by the baseline. This is also expected, as it only captures the semantic signal with no emphasis on the lexical similarity between the query and the background articles.

### Integration of relevance signals

Having noticed that the semantic matching signal is not sufficient for an effective background linking, and to address **RQ2**, we experimented with the different score and rank aggregating methods that integrate both the lexical and the semantic relevance signals. For that purpose, we adopted only the SBERT semantic model, as it exhibited the best performance among the encoder models. Table 3 presents the results of this experiment.

The results show that score aggregation methods are generally better than rank aggregation methods. More importantly, three of them, namely **SumOfScores**, **ProductOfScores**, and **MixedMinMaxScore** exhibit *statistically significant* improvement over the baseline, achieving a new SOTA for this problem. The difference between the three aggregation methods is not statistically significant though, favoring the SumOfScores method for simplicity and efficiency purposes. It is important to recall that the baseline method here denotes using solely the lexical matching scores for ranking the candidate articles. Accordingly, the significant improvement in performance here highlights the effectiveness of our proposed relevance signal integration approach.

**Table 3  nDCG@5 performance of the different lexical-semantic aggregation methods.** Results marked with asterisk (*) show a statistically significant difference over the baseline using a pairwise *t*-test with a 95% significance level.

| Aggregation method | nDCG@5 |
| --- | --- |
| SumOfScores | 0.5016* |
| MaxScore | 0.4854 |
| MinScore | 0.4736 |
| ProductOfScores | 0.5003* |
| MixedMinMaxScore | 0.5021* |
| Borda Rank | 0.4831 |
| Dowdall Rank | 0.4775 |
| Baseline | 0.4856 |

### *Potential of query-based integration of relevance signals*

While the above experiments demonstrated that the integration of the lexical and semantic relevance signals exhibited significant improvement over the SOTA baseline, it was applied to *all* queries in the same way. This is simple, but probably not optimal; we hypothesize that some queries might benefit *only* from the lexical signal, some might benefit *only* from the semantic signal, and others might benefit more from *mixing* both with different balances. To test this hypothesis and address **RQ3**, we conducted an experiment where we computed the final score of a candidate article by linear interpolation (weighted average) of the lexical and semantic scores, *i.e.,* $R(b_j, q) = (1 - \alpha) \cdot L(b_j, q) + \alpha \cdot S(b_j, q)$, where $\alpha \in [0..1]$ is the weighting factor. We changed the value of $\alpha$, per query, from 0 (no semantic signal) to 1 (no lexical signal) with an increment of 0.1. Figure 6 illustrates how many queries achieved its highest nDCG@5 score at the different values of $\alpha$. We can draw several interesting observations from the figure. First, query articles widely vary in their need for lexical and semantic relevance signals, which indicates that our simple integration is clearly sub-optimal. Second, 97 queries (about 47% of the query set) achieved their best nDCG@5 score using a pure lexical relevance signal. We can also notice from the figure that 70 queries (about 33.5% of the query set) needed more semantic than lexical signals, *i.e.,* had their best nDCG@5 scores with $\alpha > 0.6$.

We hypothesize that query articles that needed a pure lexical signal for the matching process may have reported specific *incidents* or *events* with its implications or circumstances, for which finding background articles that are written with the same vocabulary increases the chance of acquiring more knowledge on either the reported event or its direct surrounding context that is discussed in the article. Randomly investigating some of these queries, we found, for example, the query article titled "*Metro, Local Responders Would Jointly Test Radios Under New Plan*". This article provides feedback on an incident at L'Enfant Plaza Station in the United States, where a train became disabled and smoke filled the tunnel, resulting in deaths and injuries. The article illustrates how there will be a test of the equipment that was not functioning properly during the incident. Clearly, the reader of the query article would want to know more details about this specific incident that was not detailed in the query article. One of the highly-relevant background articles for

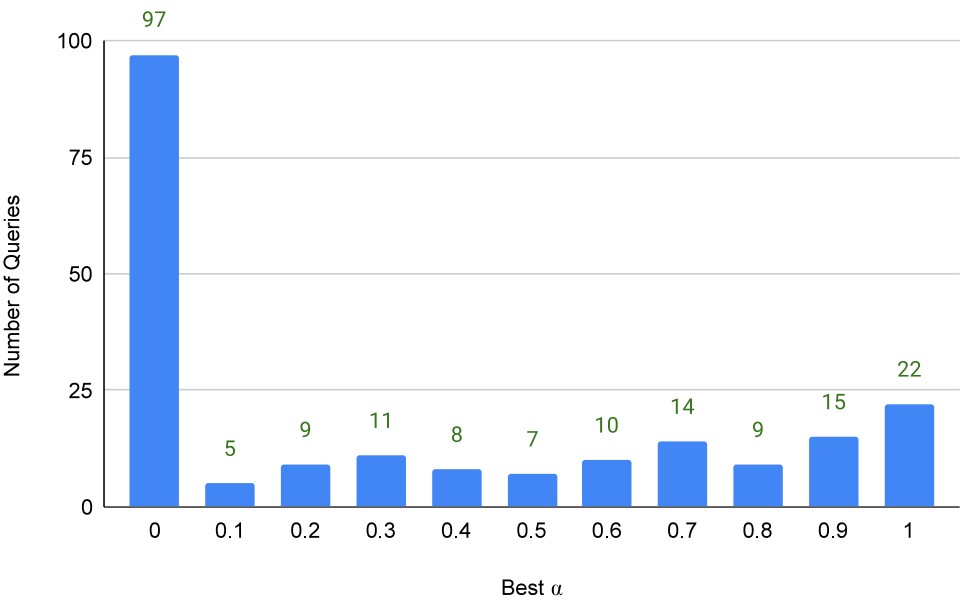

**Figure 6** Distribution of queries over the values *α* that achieve the best nDCG@5 scores per query.

this query is the article titled "*Montgomery Firefighters Find Radio-signal Blind Spots Near 2 Metro Stations*". That article mentions the exact same incident with more complementary details on why the radio signals were not functioning. It accordingly has high frequencies for words mainly used in the query article, such as "Metro", "Plaza", "firefighters", "radio", "communication", and "smoke".

On the other side, an example of the queries that needed a pure semantic signal during the matching process is the query article titled "*The Solution To Climate Change That Has Nothing To Do With Cars Or Coal*". That article specifically talks about the tropical forests near the Amazon river in Brazil and how cutting trees there had a negative effect on the climate change recently. For this article, articles that mainly talk about the dangers to the forests, the recent climate change, or either that are not necessarily around the Amazon river would increase the reader's knowledge on the subject of the main article. For instance, we found that the article titled "*The Forests of the World are in Serious Trouble, Scientists Report*" is highly relevant although it uses different vocabulary than the one used in the query, with big gaps in the frequencies of even common words. That article discusses mainly the major threats to forests of the world one by one. It hence uses words like "temperate", "boreal", and "planted", which are not at all mentioned in the query. Words like "Amazon", "Brazil", and "Climate" which mainly shape the discussion in the query article are infrequent in that candidate. Accordingly, for those type of query articles, the exact same vocabulary used in the article is not actually essential for matching the background links, rather than the actual context of the text. Presumably, using semantic relevance signals while ranking candidates for those articles ensures that the background articles are favored solely based on the context, regardless of the vocabulary being used.

Those observations introduced here need to be supported by an extensive analysis of many query articles. However, our simple analysis here sheds the light on the importance of distinguishing different types of query articles, which is a potential future direction for working on our problem.

Finally, to see how far we can go if the optimal value of $\alpha$ is predicted, we computed the overall nDCG@5 score by choosing the best $\alpha$ achieved by each query. In that case, the overall nDCG@5 reaches 0.5487, which is quite a significant increase over the SOTA baseline method (nDCG@5 value of 0.4856). Even if we restrict the values of $\alpha$ to be only 0 or 1 per query (i.e, to use either a lexical or semantic-based reranking of candidate background articles), we can reach an nDCG@5 score of 0.5262, which would still be significantly higher than the baseline performance. This shows a potential significant improvement with models that can attentively predict a better integration of the lexical and semantic signals per query. The challenge here is that we only have a very limited number of query articles (only 207) available for training and testing such a potential model. We hence leave building that model to the future research on the problem.

### Limitations

In this work, we studied how we can best represent news articles semantically for news background linking. We showed that using pre-trained encoder models, which take short fragments of text, such as sentences, as input, one could have a representation for news articles that, to a great extent, captures its semantics. It is important to highlight here though that our evaluation of the different models that we experimented with here was based mainly on the effectiveness achieved when matching news articles in the news background linking task. This task is different from other tasks that involve text matching, for which those models may have been or may be used. In other words, the conclusions that we draw here might not generalize to other tasks that also require the semantic representation of news articles or generally textual documents. Our study is also limited in the number of models involved in the comparison. We believe that it can be further improved by including other models that were introduced recently and that were mainly proposed for long text matching tasks, such as Match-ignition (*Pang, Lan & Cheng, 2021*) and CoLDE (*Jha et al., 2023*). As far as we know, though, those models were not released, thus they need to be trained from scratch for either inference or fine-tuning purposes. Accordingly, we leave this as a future work.

## IMPLICATIONS OF OUR STUDY

In this section, we outline the theoretical and practical implications of our study.

### Theoretical implications

Several studies attempted to address the news background linking problem; however, our proposed method highlights the idea of looking at the query news article as a set of subtopics that each might be expanded with background knowledge in the retrieved news articles. While we adopted a simple way to determine the subtopics that used a sliding window on the query article's consecutive paragraphs, we paved the way for a future work that may investigate other ways to better split the query article into subtopics.

Additionally, our work introduces a simple hierarchical way to obtain representations for news articles using sentence-based transformers without further training or fine-tuning the language models. This method can be leveraged by researchers to address other problems that need to represent news articles in the semantic space, such as stance detection, topic modeling, or fake news detection. It might further encourage researchers to adopt pre-trained dense encoders, in a zero-shot setting, in new downstream tasks (*e.g.*, to analyze the frequency of certain topics over time to identify trends in public interest or media coverage).

Furthermore, our study is the first to report performance on *all* query articles released for the news background linking task. This enables an evaluation framework that is both fair and more reliable. We also release our source code for both our proposed background linking approach, as well as the full-article baseline approach to support future performance comparisons or enhancements to our proposed approach.

Finally, our proposed work may further inspire researchers that work on other problems that require matching pairs of text documents. Instead of matching the whole pair of documents directly, one may consider first to match one document from the pair with passages from the other, then aggregate those matching scores. It is essential to consider the best splitting strategy given the nature of the research domain.

### Practical implications

Adding background links manually is indeed a tedious process by the authors. Therefore, we envision that our proposed automated system can be used offline by journalists to assist them in enriching their written articles with useful background links that enhance the learning experience of their readers. Moreover, enabling authors to automatically add these links can save them time and effort, allowing them to focus more on their written stories. Furthermore, our proposed system can be used online by readers who are seeking more knowledge on the content of a specific article that they do not fully understand its context.

## CONCLUSION AND FUTURE WORK

News background linking is a challenging problem that was recently introduced to the research community within the Text Retrieval Conference (TREC). In this problem, users are assumed to be reading an input query article that they do not quietly comprehend and thus seek more knowledge to aid them in the contextualization of the article's content. In this work, we proposed a zero-shot framework that can effectively find relevant news background links for input query articles. Our framework integrates the lexical matching signal between the query article and the candidate background link and the semantic matching signal between both using their dense representations. We investigated a number of Transformer-based encoder models, in a zero-shot setting with no further pre-training nor fine-tuning, to represent the news articles in the semantic space. We further investigated a number of score aggregation functions for reranking the candidate articles. We found that using a hierarchical aggregation of sentence-level representations is best for semantic matching. We further found that by simply summing the normalized lexical and semantic

matching scores of candidate background links, we achieve an effective background linking, significantly outperforming the SOTA for the problem.

We, moreover, discovered that the query articles differ in the degree to which they need the incorporation of the semantic relevance signal in the reranking process of its candidate background links. While some articles considerably benefit from the semantic signal, others are penalized. The future research direction in this problem should hence investigate how to distinguish between the different query types. One of the challenges in this direction, though, is that there is only one dataset available for this problem, with a limited number of query articles. Accordingly, there is no sufficient data for training and testing such a prediction model. Therefore, we leave building this model as a future research direction. We plan to either increase the number of queries, by annotating more articles from the Washington Post dataset or leveraging other datasets that can be adapted for news background linking.

### Funding
The authors received no funding for this work.

### Competing Interests
The authors declare there are no competing interests.

### Author Contributions
- Marwa Essam conceived and designed the experiments, performed the experiments, analyzed the data, performed the computation work, prepared figures and/or tables, authored or reviewed drafts of the article, and approved the final draft.
- Tamer Elsayed conceived and designed the experiments, analyzed the data, authored or reviewed drafts of the article, and approved the final draft.

### Data Availability
The TREC Washington Post Corpus data is available after signing an agreement with NIST at https://trec.nist.gov/data/wapost.

The code is available at GitHub and Zenodo:

- https://github.com/Marwa-Essam81/ZShotNewsBackLinking

- Marwa-Essam81. (2024). Marwa-Essam81/ZShotNewsBackLinking: Zero Shot Reranking For News Background Linking V1 (ZS_NewsLinkingV1). Zenodo. https://doi.org/10.5281/zenodo.13955936.

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
