# Peer review of "Zero-shot reranking with dense encoder models for news background linking"

_PeerJ Computer Science, doi:10.7717/peerj-cs.2534_

## Round 0.1 · original submission · Major Revisions

With respect to the reviewers’ comments and my reading of the paper, the research questions are well defined and investigated. However, the reviewers have raised several issues concerning a lack of clarity in the basic reporting and experimental design. In addition, Reviewer 2 would like to see a link to the code. Further details about submitting code as a supplementary file or via a link can be found in the authors’ instructions https://peerj.com/about/author-instructions/. I have therefore decided that the manuscript requires major revisions before it could be reconsidered for publication in the journal. Please respond, point by point, to the issues raised by the reviewers, making clear any changes in the manuscript.

Reviewer 1 ·

Basic reporting

• How did you ensure the effective integration of lexical and semantic relevance signals? Did you experiment with different weighting schemes for the integration, and if so, what criteria did you use to determine the optimal weights?
• How did you ensure the accuracy and relevance of the dense representations obtained from pre-trained models in a zero-shot setting? Were there any specific strategies employed to mitigate the absence of labeled data?
• Did you employ any techniques to split and aggregate the text to ensure comprehensive representation?
• Please elaborate on the evaluation metrics and methods used to ensure a fair comparison with the SOTA baseline? How did you validate the statistical significance of the observed improvements?
• Provide more details on the hierarchical aggregation process. How did you ensure that the aggregation method captures the semantic nuances of the news articles?
• Analyze and address the variability in the degree of need for lexical versus semantic signals for different query articles. Was there a method used to predict the optimal integration balance for each query?
• Consider the applicability of your methods to other domains? What steps would be necessary to adapt your approach to different types of text or domains?

Experimental design

Provide a more detailed explanation of the methodology, including the specific steps and algorithms used for integrating lexical and semantic signals. Include visual aids, flowcharts, or pseudocode to make the process clearer. Explain the hierarchical aggregation process with concrete examples.

Validity of the findings

Provide a more rigorous validation of the integration methods. Conduct ablation studies to demonstrate the individual and combined effectiveness of lexical and semantic signals. Compare the performance of different integration techniques and justify the choice of the final method based on empirical evidence.

Cite this review as

·

Basic reporting

The authors provided an in-depth analysis of previous approaches for news background linking and their results, mainly related to the Text REtrieval Conference.

In related work you mention some techniques were "Using the information need that was only available for some query articles", please explain more in detail what is this "information need".

The english is good overall, apart from some phrasing that could be clearer, e.g. L130-132, L515 ("causing the killing and injuring of people" is not right), L539 ("though" is not necessary), L556-557, L606 (should be "others get hurt" or are penalized)

Also you should prefer using "their" instead of "his/her", "they" instead of "he/she", and "them" instead of "him/her", it is easier to read.

Experimental design

Research questions and the problematic behind news background linking are well defined

The method is properly detailed and clear. Many approaches have been investigated and compared.

Validity of the findings

Results achieved an improvement over the state of the art. Additional research have been done to better understand the results of the experiment (why lexical matching works so well on some articles without the need for semantic matching).

The conclusion is sound and the author propose many interesting options for future works

Only one major remark: the article mention to "publicly release the source code of our methods and experiments" at least 3 times, but no link to the code has been provided

After a quick search on the internet I could find this publication on Zenodo: https://zenodo.org/records/7329399, but I am not sure if it is the right and complete code, and there are no documentation on how to use it

To make this contribution useful to other researchers in the field of computer science:
1. the code should be published to a public code repository (github, gitlab, codeberg) and/or zenodo
2. the codebase should contain a README.md with all instructions on how to install dependencies and run the code against the TREC benchmark. Ideally to be able reproduce all the results the author got in the paper, or at minima to reproduce the SOTA results of lexical + semantic matching.
3. there should be a link to the codebase in the article. Somewhere easy to find (abstract, introduction or conclusion), so that readers of the article can easily find the code (which is the most important part of this article!)

---

## Round 0.2 · Minor Revisions

Reviewer 1 is satisfied with the revision which has addressed all their concerns. Reviewer 2 has followed the code link and has several concerns. I have decided to request minor revisions to give you the opportunity to consider and respond to the feedback provided by Reviewer 2. You are not obliged to make all these changes for acceptance but some responses to the reviewer's comments are needed. PeerJ make all code accessible so code quality and reproducibility of results are important. When resubmitting please include a separate summary of the changes to the code that were made.

Reviewer 1 ·

Basic reporting

The authors have addressed all my comments

Experimental design

Experiment section is well-structured

Validity of the findings

No comment

Cite this review as

·

Basic reporting

no comment

Experimental design

The current documentation and code quality available at https://github.com/Marwa-Essam81/ZeroShotRerankingNBL is not satisfying for reproducibility in scientific research. Please address the following points:

1. In the README.md it is not clear which dataset needs to be downloaded: "you need to download first the Washington Post collection file from https://trec.nist.gov/";. Please provide the exact URL where the data can be downloaded: https://trec.nist.gov/data/wapost in the README.md

Since the process to get the WaPost data is so constraining and not guarantee to succeed ("Subject to our approval") please make it so the project can be run on a small set of example data without the need to get the WaPost data from TREC (access to this data might not be possible in the future).

2. "You need to have these dependencies in your created java project: JSOUP, Lucene V8.0". This is not sufficient for dependencies resolution. You should provide a file that describes the java project and its dependencies in a way that it can be installed automatically. You can for example provide a pom.xml file that will work with maven for example. You should also clarify on which version of java you made it work in the readme

3. You should provide the exact sequence of bash commands to run at the root of the code repository to compile the project and run it on the downloaded data. Other researchers should not have to struggle setting up your project, you should have already done it for them. e.g.

mvn package
java -jar target/ZeroShotRerankingNBL.jar

4. The code is commented, but it is not properly formatted, at a point where it is hardly readable at some places (e.g. in BackgroundLinking.java around line 66 the indentations are going all over the place). There are automated formatters integrated in most IDEs. In VSCode all you need to do is to right click in the file and click "Format document" or hit Ctrl + Shift + i in your keyboard. Eclipse and IntelliJ also have options for that.

5. There is no LICENSE or LICENSE.txt file describing the license under which your code is published. It is required to provide a license so your code can be leggaly reused. Ideally use a standard open source license such as MIT or Apache.

6. Give instructions on how to start the SQL database with the expected configuration. Ideally provide a compose.yml file so it can be started easily on any platform for reproducibility with one docker compose up command. Or you could probably use sqlite so that the SQL database is started directly by the Java program in a local file. Don't forget to also initialize the database with the right tables.

7. The IDE is showing an error in indexer.java, so not sure if it will even run

8. "An example on how to call the retrieval process is given in the main method": this should be made clear in the README.md. As previously mentioned there should be a clear step by step process to deploy the required infrastructure, load the data and run the code in the README.md.

9. What is the 1000 lines utils.py python file for? It is not mentioned in the README.md, even if there are more lines of python than java code overall in the project. If it is used please provide a pyproject.toml or requirements.txt file with the required dependencies and their version.

10. Delete the files that are not used anymore (e.g. the ReadMe.txt that have been replaced by README.md)

Validity of the findings

no comment

---

## Round 0.3 · accepted · Accept

Thank you for submitting revised a source code repository and manuscript. I appreciate the effort you've made to address Reviewer 2’s comments and find that the resource code repository is much improved. The paper and accompanying code read well and the source code repository provides useful support future research on this problem. I am happy to accept the revised manuscript for publication. Congratulations.